# HPTLC-Based Chemical Profiling: An Approach to Monitor Plant Metabolic Expansion Caused by Fungal Endophytes

**DOI:** 10.3390/metabo11030174

**Published:** 2021-03-17

**Authors:** Luis F. Salomé-Abarca, Cees A. M. J. J. van den Hondel, Özlem Erol, Peter G. L. Klinkhamer, Hye Kyong Kim, Young Hae Choi

**Affiliations:** 1Natural Products Laboratory, Institute of Biology, Leiden University, Sylviusweg 72, 2333 BE Leiden, The Netherlands; o.erol@biology.leidenuniv.nl (Ö.E.); h.k.kim@biology.leidenuniv.nl (H.K.K.); y.choi@chem.leidenuniv.nl (Y.H.C.); 2Department Molecular Microbiology and Biotechnology, Institute of Biology, Leiden University, Sylviusweg 72, 2333 BE Leiden, The Netherlands; c.a.m.van.den.hondel@biology.leidenuniv.nl; 3Plant Ecology and Phytochemistry, Institute of Biology, Leiden University, Sylviusweg 72, 2333 BE Leiden, The Netherlands; p.g.l.klinkhamer@biology.leidenuniv.nl; 4College of Pharmacy, Kyung Hee University, Seoul 02447, Korea

**Keywords:** fungal selection, metabolic differentiation, chemical diversity, antifungal activity

## Abstract

Fungal endophytes isolated from two latex bearing species were chosen as models to show their potential to expand their host plant chemical diversity. Thirty-three strains were isolated from *Alstonia scholaris* (Apocynaceae) and *Euphorbia myrsinites* (Euphorbiaceae). High performance thin layer chromatography (HPTLC) was used to metabolically profile samples. The selected strains were well clustered in three major groups by hierarchical clustering analysis (HCA) of the HPTLC data, and the chemical profiles were strongly correlated with the strains’ colony size. This correlation was confirmed by orthogonal partial least squares (OPLS) modeling using colony size as “Y” variable. Based on the multivariate data analysis of the HPTLC data, the fastest growing strains of each cluster were selected and used for subsequent experiments: co-culturing to investigate interactions between endophytes-phytopathogens, and biotransformation of plant metabolites by endophytes. The strains exhibited a high capacity to fight against fungal pathogens. Moreover, there was an increase in the antifungal activity after being fed with host-plant metabolites. These results suggest that endophytes play a role in plant defense mechanisms either directly or by biotransformation/induction of metabolites. Regarding HPTLC-based metabolomics, it has proved to be a robust approach to monitor the interactions among fungal endophytes, the host plant and potential phytopathogens.

## 1. Introduction

Natural products are considered to be the most abundant source of bioactive chemicals of all time, due, perhaps, mainly to their great chemical diversity [1]. Plants, animals, bacteria, fungi, algae, and other marine organisms could be considered as a bank of valuable metabolites to which we have turned time and time again in the search for compounds or leads as solutions in the most diverse fields such as pharmaceutics, food chemistry, agriculture and cosmetics [2]. Their choice as possible candidates for many of these applications has often been the result of ethnopharmacological information passed on from generation to generation or by the semi-synthesis and/or redesigning of molecules identified from such sources. This has resulted in the expansion of the diversity of metabolites and/or their selectivity and potency [1].

The chemical diversity of a given organism can originate in any or all of the following three sources: the organism itself, associated organisms and/or the result of interactions between the organism and associated organisms [3]. The latter encompass plant to plant communication and plant-herbivore/ fungal/ bacterial/pathogen and symbiont interactions. Among these, the interactions between plants and associated microorganisms are quite complex, leading to diverse and unpredictable metabolic outcomes [4]. Even under specific circumstances, associated bacteria or fungi could convert from beneficial to pathogens or saprophytes [4,5].

One interaction of special interest is that of fungi that colonize healthy plants intercellularly and/or intracellularly without showing conspicuous symptoms. Since their discovery, these fungi, known as fungal endophytes [6], have become one of the main sources of new bioactive molecules and many studies have revealed their potential to synthetize numerous compounds that promote plant growth and also help them to persist in their natural niches [7]. For example, several specialized metabolites such as camphotecin, podophyllotoxin, and paclitaxel were found to be fully or partially synthetized by endophytes [8,9,10]. It has been observed that endophytic interactions could play a significant role in induced defense mechanisms in their host-plants. This includes the increase in constitutive compounds and the production of new specific specialized metabolites [11].

Despite interest in endophyte research, many of their numerous alleged biological roles have not been sufficiently proved. Although the roles of fungi in host-plant-fungal interactions still remain widely unknown, it could be expected that no matter how plants and endophytes interact, at the end of the day plants have substantial influence over their metabolic production and vice versa. An example of these interactions is the case of the Convolvulaceae family and Clavicipitaceous fungi that leads to the synthesis of ergoline alkaloids by the fungi [12] and *Fusarium solani* are thought to be involved in the synthesis of camptothecin, the main alkaloid of *Camptotheca acuminate*, [8].

Similarly to the fields of biology or biochemistry, chemical ecology research has benefited from the emergence of metabolomics, an approach based on comprehensive metabolite profiling techniques. The contribution of this type of technique is not limited to its potential for acquiring more data thanks to the use of newer and more advanced technology but rather to its integration with powerful data processing and management software that offers comprehensive holistic pictures of a given situation. As a result, a holistic chemical analysis has become a must-have in the field of chemical ecology, implying the need of counting on efficient analytical tools that meet stringent requirements of sensitivity, resolution, and data robustness [13].

Among the existing analytical platforms that can meet such requirements appropriately, nuclear magnetic resonance (NMR) spectroscopy and mass spectrometry (MS) are undoubtedly the most popular in terms of broadness of metabolic coverage, reproducibility and sensitivity [14]. However, even these have their limitations and it often necessary to use combinations of techniques with either on- or off-line hyphenated analytical tools [13,15]. However, even then, there are still many limitations in the use of those analytical platforms.

Recently, high performance thin layer chromatography (HPTLC) has gained ground as an interesting alternative for chemical profiling, alone or as a complementary tool [16,17]. Its multiple advantages include its broad range of chemical profiling options [18] and enhanced detection thanks to the possibility of using numerous post-analytical chemical reagents [19], speed of analysis, and parallel analysis of samples [20]. Moreover, HPTLC has proved to be efficiently coupled to MS and NMR spectroscopies [20,21,22,23]. Additionally, its full automatization and relatively easy interpretation using multivariate data analysis have consolidated this technique as a high throughput metabolomics method, mainly in plant science studies [18,20,24,25].

In view of these promising results, HPTLC-based metabolomics was also scrutinized for its implementation to explore and characterize the role of plant fungal endophytes in the chemical diversity and defense of their host-plants. That is, not at compound identification level but at biological characterization and descriptive level. To do so, fungal endophytes were isolated from *Alstonia scholaris* (Apocynaceae) and *Euphorbia myrsinites* (Euphorbiaceae) and then chemically profiled using HPTLC. Subsequently, based on their metabolic differentiation and their correlation to their colony features, representative strains were selected for co-culture assays against fungal pathogens and plant metabolites feeding experiments. The results of these experiments were expected to reveal details of the direct participation of fungal endophytes in the host-plant defense system and their role in the host-plant metabolic diversity expansion.

## 2. Results

### 2.1. Endophyte Isolation and HPTLC Multivariate Data Analysis

The isolation of culturable fungal endophytes from *A. scholaris* and *E. myrsinites* was performed to get endophytic models to test an HPTLC-based metabolomic profiling approach to assist a non-randomized strain selection based on intrinsic metabolic and biological fungal characters. Moreover, derived from this approach, a second branch, antifungal activity enhancement of non- or poorly active fungal strains, was explored after host-plant metabolization. The general workflow can be observed in Figure 1. Then, the step-by-step results are described.

The culture of two plant species after their surface sterilization resulted in 102 fungal isolates. A first selection step of the fungal strains was performed on the basis of their macroscopic morphology. The selection yielded 33 strains with unique morphological features such as colony size, border, color, and pigment production (Appendix A). Their chemical production was profiled and monitored by high performance thin layer chromatography (HPTLC), and the extracted data was interpreted by multivariate data analysis (MVDA).

Along with the HPTLC profiles, ^1^H NMR was applied to the strain with a more distinctive HPTLC profile for its comparison. However, the ^1^H NMR spectra of the fungal extracts yielded only very weak signals in the aromatic region, which made data extraction or interpretation impossible. Conversely, the visual inspection of the HPTLC plates showed a much more conspicuous differentiation among metabolic profiles of the fungal strains. Additionally, HPTLC data was highly reproducible, with a very negligible visual variation between biological replicates, and Rf values among biological replicates (Appendix A). This metabolic screening experiment showed that the HPTLC is an optimum metabolomic platform for the differentiation of fungal endophyte strains in terms of speed and coverage of metabolites even without chemical derivatization reagents. Taken as an example, metabolic profiles did not always match morphological data, that is, even strains with very similar morphological features such as *Penicillium* and *Talaromyces* showed greatly distinctive metabolic patterns. Subsequently, a hierarchical cluster analysis (HCA) model was constructed using 18 reduced principal components (PCs) of the principal component analysis (PCA) of the HPTLC data of the fungal strains. The HCA dendrogram provided three main clusters grouped by metabolic similarity among fungal strains (Figure 2a,b). The strains are well clustered in three groups in the PCA and HCA by their metabolic profiles: group 1 (green branches) with 12 samples, group 2 (blue branches) with 39, samples, and group 3 (red branches) with 46 samples.

### 2.2. HPTLC-Fungal Growth Correlation and Co-culture Bioassays

The next step consisted in the selection of representative strains from each cluster of the HCA. With this in mind, the correlation between the fungal metabolic variation and their growth rate (colony size) was studied with unsupervised multivariate data analysis considering the colony size of each isolated strain and the HPTLC metabolic data obtained for each by an orthogonal partial least square (OPLS) analysis. The OPLS model was well validated according to the *Q*^2^-value (0.44) of the permutation test, and the *p*-value (*p* = 5.09 × 10^−10^) of the analysis of variance testing of Cross-Validated predictive residuals (CV-ANOVA) test. Thus, the model showed that almost half the metabolic variation of the fungal strains was strongly correlated to their colony size, which can be, at the same time, associated to their growth rate (Figure 3). Therefore, the biggest or fastest growing strains of each cluster were selected for the next experiments. Moreover, two strains were selected from one cluster to control whether they displayed similar biological behavior or not. The internal transcribed spacer (ITS) identification of the four selected strains is shown in Table 1.

To assess a potential direct participation of the fungal endophytes in the defense system of their host-plants, the four previously selected strains were individually challenged against four general pathogenic fungi (*Alternaria alternata*, *Botrytis cinerea*, *Colletotrichum acutatum* and *Fusarium oxysporum*) in dual co-culture assays. All the strains showed a degree of antifungal activity against at least one of the tested pathogenic fungus. The antifungal effect can be visualized as an inhibition area in the confrontation zone in the middle of the agar plate (Figure 4). Interestingly, the fungal strains from the three different groups showed totally different behaviors against each pathogen but the two strains from the same cluster (11(2) and 4/3/2) showed a similar behavior in the co-culture assays (Figure 4). The strain 5/3/2 showed the strongest and clearest antagonistic effect against the four phytopathogenic fungi (Figure 4). Conversely, the strain 11(2) showed little effect against the phytopathogens when in close contact with the pathogenic fungi (Figure 4). Interestingly, an inductive effect on the production of pigmented exudate was also observed in this strain, especially against *C. acutatum* (Figure 4).

The HPTLC analysis of extracts obtained from the confrontation and non-confrontation zones of the dual co-cultures showed an increase in constitutive metabolites, together with the induction of new compounds observed as more concentrated bands in the range of Rf 0.20–0.38 and new chromatographic bands at Rf 0.52–0.72, respectively. Taken as an example, the chromatographic profiles of 11(2) showed a clear increase in constitutive bands and the appearance of new chromatographic bands (Figure 5). This demonstrated inducible mechanisms set after fungal interactions, which allow fungal endophytes to directly participate in the defense of their host-plants.

### 2.3. Fungal Feeding Experiments and Antimicrobial Activity

To prove the potential involvement of endophytes in the chemical diversity expansion of their host-plants, the two less antifungal active strains (11(2) and EU11) from the co-culture experiments were chosen as model microorganisms. The non-active endophytic fungal strains were grown in liquid medium with and without plant metabolite pools (methanol extracts) from their corresponding host-plants. Subsequently, the chemical patterns of their mycelium and growth medium were separately examined by HPTLC, and their biological activity against *F. oxysporum* was tested.

Thus, ethyl acetate extracts of mycelium and the liquid medium were extracted individually. The HPTLC profiles from the mycelium extracts of non-fed and fed fungi were well differentiated (Figure 6). The metabolic changes were more evident in the strain 11(2) isolated from *A. scholaris* (Figure 6a). That is, several new bands in the range of Rf 0.31–Rf 0.84 were found in the mycelium extracts of 11(2) fed with its host-plant metabolites. For EU11, after feeding with its host-plant extract, one new band was clearly observed around Rf 0.50 after chemical derivatization (Figure 6b). In the case of the liquid medium extracts, there were no clear differences between controls and the extracts of fed fungi. Nonetheless, for 11(2) some weak bands, which were also found in the mycelium, were also weakly detected in liquid medium extracts.

Because of the extreme metabolic change that occurred in 11(2), the antifungal activity of both mycelium and liquid medium extracts of non-fed (control) and fed 11(2) was measured against *F. oxysporum*. The antifungal activity of the extract from fed mycelium was 12 times higher than the control mycelium extracts at 15 μg/mL (Figure 7a). The higher activity of these extracts compared to that of the control was confirmed by their lower minimal effective concentration (MEC) when compared to the control MEC value. That is, the MEC value of the fed mycelium extract was 7.9 μg/mL, meanwhile the control extract showed a MEC value between 31.7 μg/mL and 15.8 μg/mL. Conversely, the antifungal activity of the liquid medium extract of the control showed better activity than the liquid medium extract of the fed fungi (Figure 7b). It is important to note that the methanol extract of the host-plant used for fungal feeding did not show any antifungal activity against *F. oxysporum*. These results clearly showed that the new induced metabolites after host-plant metabolite feeding increased the antifungal activity of the exposed fungi as a part of a defense system involving both fungal endophytes and host-plants metabolites. Additionally, feeding plant metabolites to endophytes did not show any increase in their activity against bacteria. Thus, the metabolic stimulation of fungal endophytes by their host-plant compounds might result in a selective enhancement of their activity against other fungal microorganisms.

## 3. Discussion

Apart from their metabolic complexity, the metabolomic profiling of microorganisms has always been challenging due to the chemical noise caused by the components extracted from the culture medium [26]. This issue has been partially solved by sample cleanup procedures with specific resins like HP-20, solid phase extraction or liquid-liquid partitioning with organic solvents (ethyl acetate) [27,28]. However, for metabolomics purposes, this pre-purification might lead to the loss of important polar compounds aside from being time-consuming. If using HPTLC, these steps can be avoided, since the retention of compounds can be easily controlled by an adequate choice of the mobile phase and detection of metabolites can be selected by appropriate chemical derivatization. Thus, by using different HPTLC systems it is possible to retain extremely polar and non-polar compounds of the samples or even with the use of different stationary phases. Besides, various derivatization chemical reagents such as anisaldehyde or vanillin with sulfuric acid are available to visualize any group of metabolites on the plates while suppressing unwanted chemical noise. Moreover, the single use nature of the plates avoids a possible build-up of the remaining metabolites as in some high performance liquid chromatography (HPLC) columns, as it could be the case for liquid chromatography/gas chromatography-mass spectrometry (LC/GC-MS) analyses. Additionally, several samples can be running in a single analysis, and the use of silica gel allows for the combination of diverse solvents to achieve good chromatographic separation. Furthermore, the problem caused by large variation of Rf values with common low-resolution on TLC, which eventually causes problems of data reproducibility, could be overcome by automatizing all the experimental steps of HPTLC, especially sample application and plate development [29]. Not only hardware controlling processing but also data processing platforms could reduce the variation of Rf values by applying several algorithms for base-line correction, gamma correction and band alignment [24,30,31] as well as normalization of band intensity in HPTLC chromatograms [13,18].

In fact, the power of HPTLC for chemical monitoring has been previously demonstrated with many successful results for the differentiation of plant extracts, being the method of choice to confirm the identity of botanical extracts for their quality control, as fingerprinting and metabolomics-based platforms [32]. Indeed, the applications are not limited to plants but also medicinal mushrooms and their metabolite pools have been successfully profiled [33], even this metabolic profiling showed to be of high potential for mushroom quality control [34]. Additionally, metabolomic data obtained from HPTLC has been also associated with on-line bioactivity such as antioxidant [35], or determination of antidiabetic compounds [36]. This also proved to be the case in this study, in which chromatographic profiles of the methanol extracts of endophytes represented the metabolic production of the endophytic fungi, and yielded numerous chromophore metabolites, representing thousands of pixels that could be used as data for MVDA [25]. Subsequently, this data was normalized and able to distinguish and group different fungal strains unequivocally by MVDA. That is, the PCA and subsequent HCA analysis of the endophytic strains according to their chemical similarity resulted in three clusters. Thus, the strains grouped in each score cluster were clustered because of their chemical similarity.

Another advantage of HPTLC data is that it can be correlated online and offline with biological data such as antimicrobial, antioxidant or enzymatic activities, among others [37,38,39]. In this study, the HPTLC data was shown to correlate well with growth rate of the fungal strains (*Q*^2^ = 0.44, *p* = 5.09 × 10^−10^). The correlation showed that the larger the colony size, the greater its metabolic production and accumulation. This could be the result of the experimental conditions, that is, if the fungal colonies grow faster, they will fully fill the space available in the petri dish, thus entering a stationary growth phase, in which they focus on metabolic production rather than on growth [40]. Even if this looks like an artificial condition, one should take into account that when an endophyte is living inside an organism, their growth is also limited by space, as they compete with other microorganisms, and the host-plant defense mechanisms. Thus, they are also forced by this limitation to produce several metabolites for their adaptation and survival [8]. Hence, growth rate was determined to be a good selection parameter for further experiments.

Therefore, if metabolic diversity is correlated to the survival chances of an organism, then the faster growers should be chosen as models for potential host-plant defense experiments. In the co-culture experiments, especially the strain 5/3/2 showed clear antifungal effects against general pathogenic fungi. According to these results, some endophytic fungi might function as an autonomous defense system by producing their own metabolites to directly fight fungal pathogens. Conversely, the strains 11(2) and 4/3/2 showed little or no antifungal effects against general pathogenic fungi. Nonetheless, an increase in their pigment production during their development and especially during their interaction with the fungal pathogens was observed. When their different co-culture zones were analyzed by HPTLC, an increase in constitutive metabolites was observed. New chromatographic spots also demonstrated the production of new metabolites that resulted from the fungal interaction. This strongly suggests the activation of an induced defense system in fungi, which could be mediated either by de novo synthesis of defensive metabolites or an extreme increase in the synthesis of constitutive metabolites [41]. Even so, the increase in these metabolites could not necessarily fight against host plant attackers. However, at this point, HPTLC proved to be efficient for the detection and tracking of the expansion of the chemical space resulting from the induced chemical defense mechanism enhanced by fungal interactions.

Another aspect of endophyte-host plant interaction to scrutinize is the ability of endophytes to communicate with their host plants. Several plant species have been shown to affect the chemical communication between microorganisms [9,42,43,44,45,46]. Thus, it is conceivable that fungal endophytes could influence the chemical composition of their host-plants. This could occur via the new production or biotransformation of the host-plant metabolites by the endophytes or vice versa. The effect of such interactions would also affect the outcome of the host-plant ecological interactions with external microorganisms and herbivores [47,48]. Thus, the antifungal activity of the relatively non-active endophytic fungal strains might be enhanced by their host-plant metabolites. In our study, this was confirmed as new bands were observed in the chromatogram of extracts of mycelium of endophytic strains after being fed with their host-plant methanol extracts, accompanied by the increase in their antifungal activity against *F. oxysporum*.

On the other hand, the HPTLC profiling of the liquid medium extracts showed almost no changes in their composition and the liquid medium extracts of the fed fungi showed lower antifungal activity against *F. oxysporum* than the control extracts. This proved that the metabolization of plant compounds happens inside the fungal cells providing a possible explanation for the relatively lower antifungal activity exhibited by the liquid medium extracts of fed fungi. That is, when the fungal cells metabolize plant metabolites they probably focus more on internal metabolism rather than excreting metabolites. The newly synthesized metabolites can be then stored in the cells and released in the event of physical fungus to fungus contact, as occurs during mutual intermingling and inhibition (barrage formation) [49]. Thus, because they are grown as monocultures, the new metabolites are detected only in mycelium extracts, and could be detected in liquid medium extracts probably when they are eventually released in presence of other fungi (liquid dual co-culture).

In this way, the results and hypothesis obtained from HPTLC-based metabolomics and fingerprinting were confirmed by the increase in the antifungal activity of extracts from fungi fed with their host-plant extracts. Extending this to the present study, controlling the pool of endophytes (microbiome in plants) could be a strategy for the transformation and expansion of plant metabolite diversity to obtain potential bioactive molecules, which could be efficiently monitored by HPTLC-based metabolomics. Based on these results, it can be confirmed that fungal endophytes can work as an autonomous direct defense system, or as an indirect defense system activated by plant-host metabolites or by biotransformation of plant metabolites. Thus, HPTLC was proved as a good analytical platform to handle fungal metabolomes, even during fungus-to-fungus interactions. The metabolic produced data by HPTLC can be efficiently approached by MVDA, which helps to select and design biological experiments involving fungal microorganisms and their interactions. Moreover, this data can be lastly complemented by off-line or on-line MS and NMR analyses for metabolite identification [25].

Mass spectrometer is often hyphenated with HPTLC as an efficient tool for metabolic identification, which has been employed for the investigation of nucleobases in two traditional Chinese medicinal mushrooms and flavonoids of medicinal plants by HPTLC–MS/(MS^n^) [50,51]. In the case of NMR, the integration still remains in off-line mode and all applications are very limited when compared with HPTLC-MS due probably to the inherent low sensitivity of NMR. However, recently, ^1^H NMR was well integrated to HPTLC at a submicromole level of detection [52]. Thus, it is assumed that the potential of HPTLC-based chemical profiling techniques would be strengthened by further combination with post analytical platforms, that consequently open the path to get a deeper understanding of fungal interactions at both biological and metabolite levels.

## 4. Materials and Methods

### 4.1. Fungal Endophyte Isolation

Plant material consisted of leaves from *Alstonia scholaris* and stems and leaves from *Euphorbia myrsinites*. All samples were collected in the Botanical Garden (Hortus Botanicus) of Leiden University, Leiden, The Netherlands. Whole branches and aerial parts of *A. scholaris* and *E. myrsinites* respectively were manually collected and the severed edges were sealed with parafilm. At the laboratory, 5 cm of the branch edges were cut to remove exogenous microorganisms and then re-sealed with wax. For surface sterilization, the plant material was washed with tap water to remove dust particles and then subsequently sonicated for 1 min and immersed for another minute in 0.01% of Tween-20 solution. The samples were then rinsed thrice with sterile water and soaked in NaOCl (5%). After thorough rinsing (×3) with sterile water they were soaked in 2.5% Na_2_S_2_O_3_ solution for 10 min. After rinsing with sterile water (×3) they were soaked again in aqueous ethanol (75%) during 3 min (for branch samples). Finally, the samples were rinsed three times with sterile water and dried on sterile paper towels. To confirm the efficacy of the surface sterilization, 100 μL of the final rinsing water were plated on potato dextrose agar (PDA), V8 plus agar (V8), and complete medium plus agar (CM) plates. The agar plates were incubated at 26 °C in the dark, and the absence of any microbial growth was monitored over 3 days.

The sterilized plant material was cut with sterile scissors in approximately 1 cm^2^ squares, placed on PDA, V8, and CM plates and incubated at 26 °C in the dark. After 3 days the fungal colonies that started to grow at the cut edges of the samples were picked up by tapping them with a sterile straight needle and inoculated in the same media where they were able to grow. Afterwards, in order to increase the purity of the individual colonies, the isolates were streaked on agar plates and the isolated colonies were re-inoculated and streaked again to increase the purity of the colony. Finally, the purity of the colony was also checked microscopically.

### 4.2. Endophytes-Fungal Pathogens Co-culture Assays

To produce fungal plugs, mature plates were filled with 25 mL of sterile physiological solution (PS), and rubbed with a sterile cotton swab to transfer the mycelium with spores to the PS. The liquid containing the fungal structures was filtered through two layers of sterile miracloth and transferred into sterile 50 mL conical tubes. The volume was adjusted to 30 mL with PS, vortexed for 1 min and centrifuged at 4000 rpm for 10 min. This process was repeated three times. The supernatant was removed and the pellet was re-suspended in 10 mL of sterile PS. These stock solutions were individually inoculated in the center of a PDA plate with a sterile toothpick. The plates were incubated for 3 days and the plugs were taken from the margin of the colony.

Based on their chemical grouping, 4 strains were selected to be tested against four general plant pathogenic fungi (*Fusarium oxysporum*, *Botrytis cinerea*, *Alternaria alternata*, and *Colletotrichum acutatum*). The time that each strain took to reach the center of the plate was previously determined and the inoculation time for pathogens and endophytes was adjusted to make sure that both microorganisms reach the middle of the plate simultaneously. For confrontation bioassays, 92 mm agar plates were filled with 25 mL of PDA medium and after solidification a 7 mm agar plug of the pathogen was placed on one side of the dish at 1 cm from the plate edge. The plug with endophytic mycelium was placed opposite the pathogen plug also at 1 cm from the plate border. When using an agar plug resulted in over sporulation all over the plate, the fungi were directly inoculated with a toothpick taken from the border of the three days old colony. The time for the pathogens to reach the middle of the plate was: *B. cinerea* 3 days, *F. oxysporum* 4 days, *A. alternata* 5 days, and *C. acutatum* 25 days. The time for endophytes to reach half of the plate was: 11(2) 17 days, 4/3/2 17 days, 5/3/2 17 days, and EU11 12 days. The fungi with slower growth rates were inoculated in the plate first and the second microorganism was placed on the point where both of them had the same days left to reach the center of the plate. For example, 5/3/2 was placed first and 3 days before it reached the middle of the plate, the agar plug of *B. cinerea* was inoculated on the plate. Pictures of the interactions were taken at the moment both of the microorganisms reached the middle of the plate. All the experiments were carried out at 26 °C in the dark.

Plates were processed 48 h after both microorganisms reached the confrontation zone, to increase the amount of metabolite production. The plates were divided in 5 zones: the external area of the pathogen and endophyte fungus corresponded to the area of the colony closest to the wall of the plate. The inner area was limited to 10 mm from the border of the colony to the center of the colony of both pathogens and endophytes. The confrontation zone was in the middle of the plates and showed growth inhibition effects. For sample extraction, the selected zones were manually sliced into 4 mm^2^ cubes. The samples were transferred to 50 mL conical tubes. Subsequently, 25 mL of EtOAc were added to the tubes and sonicated for 30 min. The resulting extracts were separated and dried with anhydrous sodium sulfate. After concentration with a rotary evaporator, the samples were transferred to 1.5 mL glass vials and taken to total dryness with a SpeedVac (Labconco, Kansas City, MO, USA). In the case of the fungal exudates, they were directly dissolved in EtOAc. The samples were stored at −80 °C for further analysis.

### 4.3. Plant Extracts Feeding Experiments

The two fungi that showed least activity against the pathogenic fungi were selected for feeding experiments with methanol extracts of their host-plant. The methanol extracts from *A. scholaris* and *E. myrsinites* were prepared by sonicating 20 g of dried material with 200 mL of methanol for 30 min. The extracts were dried with a rotary evaporator and stored at −80 °C for the experiments. For spore production, the mycelium of the endophytes was grown in complete medium at 26°C for 1 to 3 days depending on the strain. The matured plates were filled with 25 mL of sterile physiological solution (PS) and rubbed with a sterile cotton swab to transfer the mycelium with spores to the PS. The liquid containing the fungal structures was filtered through two layers of sterile miracloth and transferred to 50 mL sterile conical tubes. The volume was adjusted to 30 mL with PS, vortexed for 1 min and centrifuged at 4000 rpm for 10 min. This process was repeated three times. The supernatant was discarded and the pellet was re-suspended in 10 mL of sterile PS. The inoculation solution was prepared by measuring the concentration of spores/ml with a cell counter (Bio-Rad, Veenendaal, The Netherlands) and adjusting it to 5 × 10^7^ spores/mL. One milliliter of the inoculation solution was added to 49 mL of Czapek broth to reach a final spore concentration of 1 × 10^6^ spores/mL. The fungi were grown at 26 ± 0.5 °C on a shaker at 150 rpm for 7 days after which the methanol extracts were re-suspended in sterile milliQ water and sonicated for 15 min. This procedure was repeated twice. The resulting methanol extracts were individually added to the medium, reaching a final concentration of 250 μg/mL. After a 7-day incubation, the cultures were filtered to separate the mycelium from the liquid medium. 

The mycelium samples were flash-frozen in liquid nitrogen and freeze-dried. The dried mycelium was consecutively sonicated with EtOAc (10 mL, 3 times) for 30 min. When the liquid medium was filtered, it was immediately poured into a separator funnel and extracted with 3 portions of 50 mL of EtOAc. The ethyl acetate extracts were dried with anhydrous sodium sulfate, filtered and concentrated with a rotary evaporator (Büchi, Flawil, Switzerland) before taking total dryness. The samples were stored at −80 °C for further chemical analysis.

### 4.4. ^1^H NMR Analysis

Ten mg of dried methanol extracts were dissolved in 1 ml of CH_3_OH-*d*_4_ containing 3.93 mM hexamethyldisiloxane (HMDSO) as an internal standard followed by 5 min of ultrasonication. All the solutions were centrifuged at 13,000 rpm for 10 min, and 300 µL of the supernatant were transferred into 3 mm-NMR tubes. The ^1^H NMR analysis was done with an AV-600 MHz NMR spectrometer (Bruker, Karlsruhe, Germany), operating at a frequency of 600.13 MHz. For internal locking, CH_3_OH-*d*_4_ was used. All ^1^H NMR consisted of 64 scans requiring 10 min and 26s as acquisition time using the parameters: 0.16Hz/point, pulse width (PW) = 30° (11.3 µs), and relaxation time of 1.5 s. A pre-saturation sequence was used to suppress the residual water signal using low power selective irradiation at H_2_O frequency during the recycle delay. The FIDs were Fourier transformed with exponential line broadening of 0.3 Hz. The resulting spectrums were manually phased and baseline corrected and calibrated to HMDSO at 0.06 ppm using TOPSPIN V. 3.0 (Bruker BioSpin, Rheinstetten, Germany).

### 4.5. High Performance Thin Layer Chromatography

All extracts were re-dissolved in sufficient amounts of their original extraction solvents (EtOAc or MeOH) to reach a concentration of 1 mg/mL and applied on silica gel HPTLC plates (20 × 10 cm, F254) purchased from Merck (Darmstadt, Germany). A CAMAG-HPTLC system consisting of an automatic TLC sampler (ATS4), derivatizer (version 1.0 AT), TLC plate heater (version III), and TLC visualizer (CAMAG, Muttenz, Switzerland) was used. For sample application, 15 μg of EtOAc extracts were spotted in 6 mm bands. On each plate, 16 bands were applied at 10 mm from the bottom and 20 mm from the lateral edges of the plate with 8 mm distance between bands. A pool of all of samples was applied on each plate at the right edge of the plate as a quality control sample for normalization purposes. The samples were separated with a mobile phase of chloroform-acetone-formic acid (8.5:0.65:0.85, *v/v/v*) and developed for 75 mm from the application point. The chamber was pre-saturated for 25 min. Developed HPTLC plates were sprayed with 2 mL of anisaldehyde-H_2_SO_4_. Subsequently, the plates were placed on a TLC plate heater at 100 °C for 3 min. Images of the derivatized plates were recorded using a TLC visualizer at 366 nm before and after derivatization.

### 4.6. Microorganisms

*Fusarium oxysporum* f. sp. *Lycopersici* was obtained from the collection of the molecular and microbiology laboratory of Leiden University. *Botrytis cinerea* was kindly provided by Dr. Jan van Kan [53]. *Alternaria alternata* (CBS 102.47) and *Colletotrichum acutatum* (CBS 112760) strains were obtained from the collection of the Westerdijk Fungal Biodiversity Institute. The bacteria used in the study: *Staphylococcus aureus* (CECT976) and *Bacillus cereus* (NCCB 75009) and gram negative bacteria *Escherichia coli* (DH5-Promega). *Pseudomonas putida* (NCCB26044), *Pseudomonas fluorescens* and *Pseudomonas viridiflava* were kindly provided by Dr. Paolina Garbeva [54].

### 4.7. Antibacterial Activity

The broth microdilution method was used to determine the minimal inhibitory concentration (MIC) of the tested triterpenes according to the Clinical Laboratory Standards Institute guideline. The strains were inoculated on Mueller-Hinton agar plates and incubated overnight at 37 °C. From the overnight cultures, a single colony was used to inoculate 10 mL of Mueller-Hinton broth (MHB) and incubated at 37 °C with constant shaking (150 rpm). The volume of the bacterial suspensions was further adjusted with the addition of MHB to 0.5 of turbidity of the McFarland scale (106 CFU/mL). Meanwhile, the ethyl acetate extracts were dissolved in DMSO and diluted two-fold to reach final concentrations in the well ranging from 512 µg/mL to 16 µg/mL. Finally, each well was inoculated with 50 µL of the 0.5 McFarland bacterial suspensions and incubated for 24 h at 30 °C. The final concentration of DMSO in the well was 5%, which was also used as a negative control. Spectinomycin at 100 µg/mL as final concentration in the well was used as a positive control. The bacterial growth was measured by optical density at 600 nm in a well microtiter plate reader (SPARK 10M, TECAN, Giessen, The Netherlands). The MIC value was defined as the lowest concentration of a compound that completely inhibited the bacterial growth at 24 h. All experiments were performed in triplicate.

### 4.8. Antifungal Activity and Minimum Effective Concentration (MEC)

#### 4.8.1. Spore Production

The extracts from mycelium 11(2) fed with *A. scholaris* metabolites and without feeding were challenged with *Fusarium oxysporum*. To produce *F. oxysporum* spores, complete medium plates were inoculated with fungal suspensions and incubated at 28 °C for two weeks. After this, the plates were filled with 25 mL of sterile physiological solution (*Ps*) and rubbed with a sterile cotton swab to transfer the mycelium and spores to the *Ps*. This liquid was removed with a sterile pipette and filtered through two layers of sterile miracloth, and transferred into 50 mL sterile conical tubes. The volume was adjusted to 30 mL with sterile water, vortexed for 15 s and then centrifuged at 4000 rpm for 10 min (3×). The supernatant was discarded and the pellet was re-suspended in 30 mL of sterile *Ps*. The spore concentration per milliliter of solution was quantified with a cell counter (TC20^TM^, Bio Rad, Kaki Bukit, Singapore).

#### 4.8.2. Minimum Effective Concentration (MEC)

The minimal effective concentration (MEC) values of the extracts were determined using the microdilution method, based on the Clinical and Laboratory Standard Institute guidelines [55] and on previously reported methods [56] with some modifications. Briefly, *Fusarium oxysporum* spores were inoculated in 2× minimal medium contained in 96 well plates. The final spore concentration in the well was 2.5 × 10^5^ spores/mL. To reach the germination state, spores were incubated for 6 h at 28 °C. When the spores had germinated, the extracts were added in a mixture of medium and DMSO. The final concentration of DMSO in the well was 1% (*v/v*) in a final volume of 200 μL. After 24 h incubation at 28 °C optical density (OD) values of the plate suspensions were measured at 600 nm with a multi-well plate reader (SPARK 10M, TECAN, Giessen, The Netherlands). The MEC was defined as the lowest drug concentration that produced a decrease in the total fungal growth with respect to the growth control. The concentration of the tested extracts ranged from 500 μg/mL in two fold dilutions to 7.9 μg/mL. The control sample consisted of spores of *F. oxysporum* growing in 2× minimal medium. The negative control consisted of fungal spores growing in the medium with 1% of DMSO (*v/v*) and the blanks consisted of medium with 1% of DMSO and the extract of the corresponding treatment. All MEC determinations were done in quadruplicate.

#### 4.8.3. Fungi Identification

The molecular identification of the selected fungal strains was performed by DNA sequencing. A spore solution of each strain was used to inoculate potato dextrose broth (PDB) and the fungal strains were grown for 3 days at 26 °C with constant shaking (150 rpm). The mycelium was then filtered through sterile miracloth. The fungal biomass was transferred to a 2 mL micro-tube and the following procedure was performed based on a previously described method [57]. One gene region was amplified using the amplification primers listed in Table 2. The PCR mixture with a total volume of 50 μL consisted of 27.5 μL of water, 10 μL of 5× Phire buffer, 8 μL of dNTP’s, 2 μL of genomic DNA, 1 μL of each primer at 20 pmol/μL, and 0.5 μL of Phire polymerase. The temperature used for the PCR was one cycle of denaturalization at 98 °C for 5 min, followed by 40 cycles of denaturalization at 98 °C for 5 s, alignment at 58 °C for 5 s, and extension at 72 °C for 30 s. Finally, a last extension step at 72 °C during 10 min was performed and the PCR products were cooled down to 4 °C for 1 min. Subsequently, the PCR products were separated on a 2% agarose gel by electrophoresis at 110 V. The separated bands were extracted and purified using a GeneJET Gel Extraction kit (ThermoScientific, Wilnus, Lithunia). The sequencing of the band products was outsourced to Macrogen (Leiden, The Netherlands). The sequences were compared with those of the GeneBank database of the National Center of Biotechnology Information by Blast analysis. The accession numbers are listed in the Appendix A.

### 4.9. Data Processing and Statistical Analysis

Data from HPTLC images at 366 nm were extracted using the rTLC software [24]. The dimensional extraction parameters were the same as those used for sample application. The pixel width for data integration was 128 and a parametric warping algorithm was applied to overcome Rf shifts. The values from the grey channel were normalized to those of the QC samples on each plate. The normalized data was analyzed by multivariate data analysis using SIMCA P (v.15.0.2, Umetrics, Umeå, Sweden). The score plots of the principal component analysis (PCA) were used to perform a hierarchical cluster analysis (HCA) to find clusters based on the chemical profile of samples. For the HCA, a ward linkage criterion was used to create the dendrogram tree. An orthogonal partial least square (OPLS) model was used to detect possible correlations between the chemical composition of the strains and their colony size (growth). The unique variance scaling method was used for scaling. The OPLS model was validated based on its *Q*2 and *p*-values obtained from the permutation and CV-ANOVA tests, respectively.

For the antifungal activity, the OD values of the blanks were subtracted from the values of the corresponding treatment to get the Δ-OD values of each well. The antifungal activity was determined by introducing the measured Δ-OD values in the formula: (100* Δ-ODt/Δ-Dgc) - 100, where 100 is 100 percent of fungal growth, ODt is the delta of the optical density of the tested treatment, and ODgc is the delta of the optical density of the growth control. The values were averaged (*n* = 4) and their standard errors were calculated and plotted using Microsoft Excel 2016.

## Figures and Tables

**Figure 1 metabolites-11-00174-f001:**
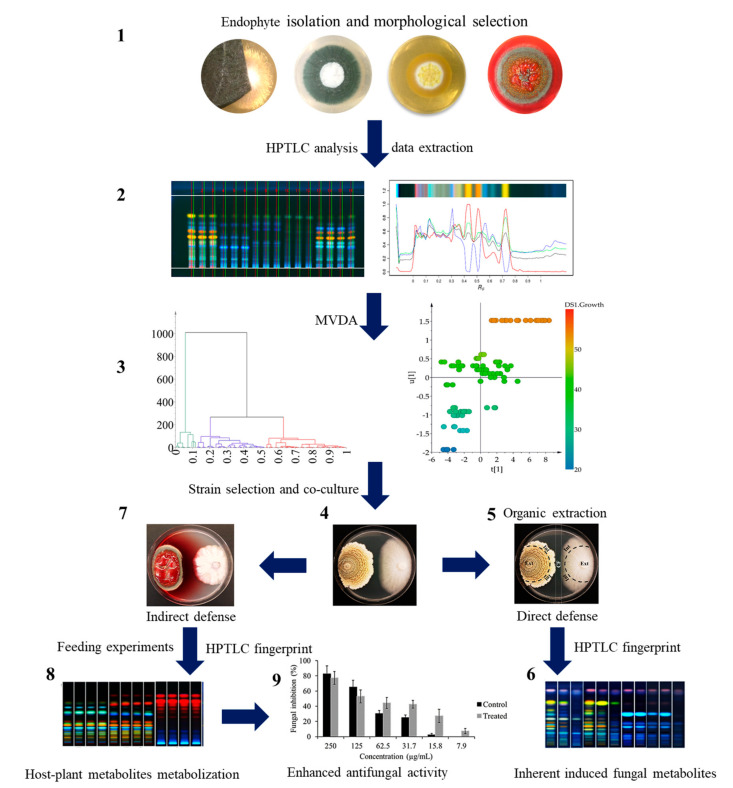
HPTLC-assisted selection of bioactive fungal endophytic strains. (**1**), Fungal isolation on solid medium and strain selection based on morphological colony features. (**2**), High performance thin layer chromatography (HPTLC) analysis, data extraction and curation. (**3**), Multivariate data analysis (MVDA) from HPTLC analysis, correlation to fungal characters (colony size), and strain selection based on chemical and biological correlation. (**4**), Co-culture bioassays of selected fungal strains. (**5**), Determination and selection of fungal endophytes with direct defensive (antifungal) capability against fungal pathogens. (**6**), HPTLC fingerprint analysis from different zones of co-cultures to determine the involvement of constitutive and/or induced metabolites on fungal defense. (**7**), Selection of fungal endophytes without antifungal effects. (**8**), Feeding host-plant metabolites to no active fungal endophytes and HPTLC fingerprinting analysis to monitor new potential metabolic production. (**9**), Determination of the fungal activity enhancement of fungal endophytic extracts after host-plant metabolization. The error bars indicate standard error of the average antifungal percentage values (*n* = 4).

**Figure 2 metabolites-11-00174-f002:**
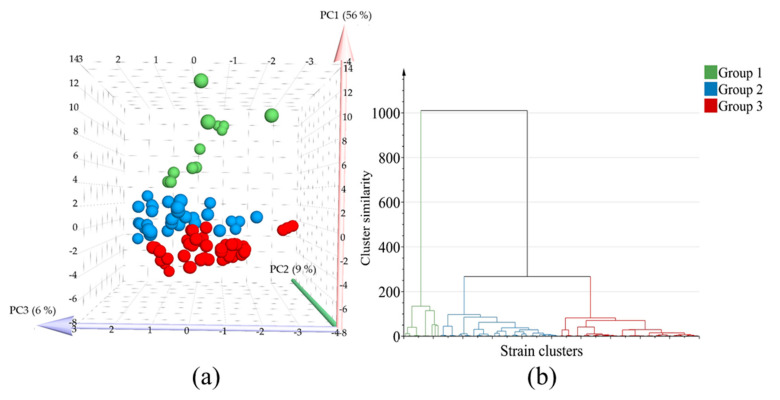
Multivariate data analysis using high performance thin layer chromatography (HPTLC) analysis data of methanol extracts of fungal endophytes grown on potato dextrose agar. (**a**), Score plot of principal component analysis using PC1, PC2 and PC3, in which three groups based on hierarchical cluster analysis (HCA) were marked with blue, green and red colors. (**b**), hierarchical cluster analysis using 18 principal components. X axis is conformed of strain names clustered by HCA and Y axis indicates the loss in within cluster similarity, that is, variance increase when clusters are merged. Thus Y axis is dimensionless.

**Figure 3 metabolites-11-00174-f003:**
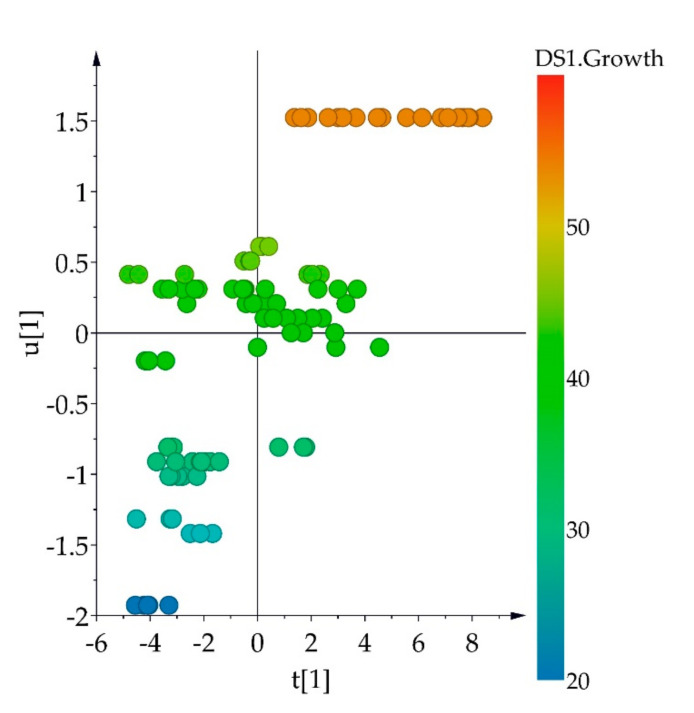
Orthogonal partial least squares analysis of high performance thin layer chromatography (HPTLC) dataset of methanol extracts of fungal endophytes grown on potato dextrose agar and their colony size. t[1] = PC1 of X-variables of HPTLC data; u[1] = PC1 of Y-variable (colony size).

**Figure 4 metabolites-11-00174-f004:**
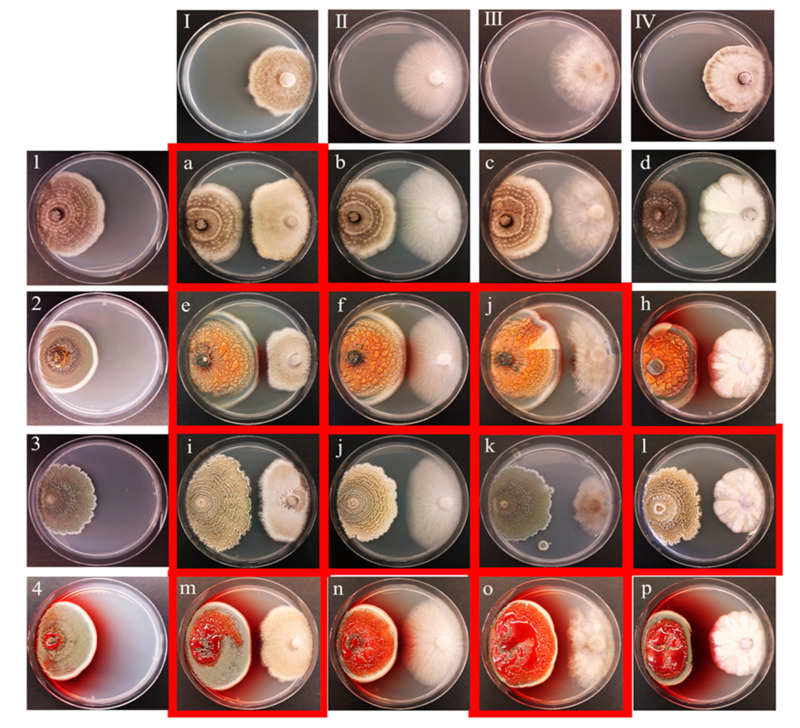
Co-culture experiments of four selected fungal endophytes and four fungal pathogens. Pathogens: (**I**), *Alternaria alternata*, (**II**), *Fusarium oxysporum*, (**III**), *Botrytis cinerea*, (**IV**), *Colletotrichum acutatum*. Endophytes: (**1**), EU11, (**2**), 4/3/2, (**3**), 5/3/2, (**4**), 11(2). Co-culture: (**a**), EU11 vs *Alternaria alternata*, (**b**), EU11 vs *Fusarium oxysporum*, (**c**), EU11 vs *Botrytis cinerea*, (**d**), EU11 vs *Colletotrichum acutatum*, (**e**), 4/3/2 vs *A. alternate*, (**f**), 4/3/2 vs *F. oxysporum*, (**g**), 4/3/2 vs *B. cinerea*, (**h**), 4/3/2 vs *C. acutatum*, (**i**), 5/3/2 vs *A. alternata*, (**j**), 5/3/2 vs *F. oxysporum*, (**k**), 5/3/2 vs *B. cinerea*, (**l**), 5/3/2 vs *C. acutatum*, (**m**), 11(2) vs *A. alternata*, (**n**), 11(2) vs *F. oxysporum*, (**o**), 11(2) vs *B. cinerea*, (**p**), 11(2) vs *C. acutatum*. The time for the pathogens to reach the middle of the plate was: *B. cinerea* 3 days, *F. oxysporum* 4 days, *A. alternata* 5 days, and *C. acutatum* 25 days. The time for endophytes to reach half of the plate was: 11(2) 17 days, 4/3/2 17 days, 5/3/2 17 days, and EU11 12 days. The fungi with slower growth rates were inoculated in the plate first and the second microorganism was placed on the point where both of them had the same days left to reach the center of the plate. Co-cultures in red squares show clear antifungal activity.

**Figure 5 metabolites-11-00174-f005:**
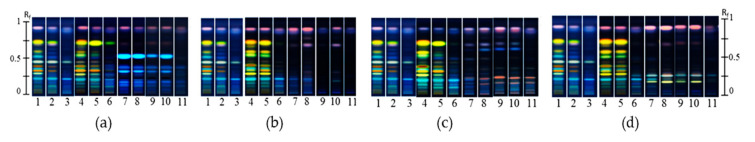
High performance thin layer chromatograms of the co-culture of the 11(2) strain and four fungal pathogens. (**a**), 11(2) against *Alternaria alternata*; (**b**), 11(2) against *Botrytis cinerea*; (**c**), 11(2) against *Colletotrichum acutatum*; (**d**), 11(2) against *Fusarium oxysporum*. In all plates, lane 1: internal zone of the colony control of 11(2), lane 2: external border of the 11(2) colony control, lane 3; peripheral medium around the 11(2) colony control: lanes 4/5: internal and border side of the 11(2) colony in co-culture conditions, respectively; lane 6: confrontation zone extract between the 11(2) strain and the corresponding pathogen: lanes 7/8 border and internal side, of the pathogen under co-culture conditions, respectively; lanes 9–11 peripheral medium, border and inner side extracts of the fungal pathogen control respectively.

**Figure 6 metabolites-11-00174-f006:**
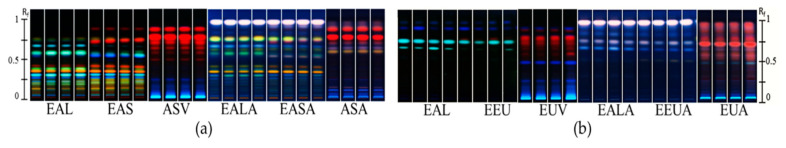
High performance thin layer chromatography analysis of mycelium extracts, visualized with ultra violet (UV) light at 366 nm with or without derivatization with sulfuric-anisaldehyde reagent. (**a**), Ethyl acetate extracts of mycelium of 11(2) fed with *Alstonia scholaris* methanol extract; EAL: mycelium extracts of 11(2) without extract feeding (negative control); EAS, mycelium extracts of 11(2) fed with methanol extract of *A. scholaris*; ASV: ethyl acetate liquid-liquid extracts of *A. scholaris*; EALA; mycelium extracts of 11(2) without extract feeding (negative control) after derivatization with sulfuric-anisaldehyde (Sa); EASA: mycelium extracts of 11(2) fed with methanol extract of *A. scholaris* after derivatization with Sa; ASA: ethyl acetate liquid-liquid extracts of *A. scholaris* after derivatization with Sa; (**b**), Ethyl acetate extracts of mycelium of EU11 fed with *Euphorbia myrsinites* methanol extract; EAL: mycelium extracts of EU11 without extract feeding (negative control); EEU: mycelium extracts of EU11 fed with methanol extract of *E. myrsinites*; EUV: ethyl acetate liquid-liquid extracts of *E. myrsinites*; EALA: mycelium extracts of EU11 without extract feeding (negative control) after derivatization with sulfuric-anisaldehyde (Sa); EEUA: mycelium extracts of EU11 fed with methanol extract of *E. myrsinites* after derivatization with Sa; EUA: ethyl acetate liquid-liquid extracts of *E. myrsinites* after derivatization with Sa.

**Figure 7 metabolites-11-00174-f007:**
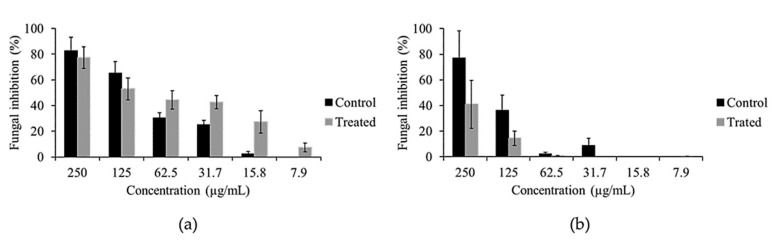
Antifungal activity comparison of mycelium and liquid medium ethyl acetate extracts of 11(2) with and without feeding with methanol extract of *Alstonia scholaris* against *Fusarium oxysporum*. (**a**), Antifungal activity of the ethyl acetate extracts of mycelium of 11(2) fed with *Alstonia scholaris* metabolites (Treated) and without feeding (control). (**b**), Antifungal activity of the ethyl acteate extracts of liquid medium of 11(2) fed with *Alstonia scholaris* metabolites (Treated) and without feeding (Control). The values represent the average of the antifungal activity (*n* = 4), and the bars represent the standard error.

**Table 1 metabolites-11-00174-t001:** Molecular identification of four endophytic fungal strains.

Figure	Matched Identity	Primer	Query (%)	Similarity (%)
EU/PDA/ST/11GPC ^a^	*Colletotrichum godetiae*	Forward	88	99.82
	*Colletotrichum acutatum*	Forward	88	99.82
	*Colletotrichum acutatum*	Forward	88	99.82
EU/PDA/ST/11GPC	*Colletotrichum acutatum*	Reverse	94	99.01
	*Colletotrichum acutatum*	Reverse	94	99.01
	*Colletotrichum acutatum*	Reverse	94	99.01
AS/PDA/5/3/2 ^b^	Uncultured ascomycete	Forward	92	99.85
	*Penicillium crustosum*	Forward	92	99.85
	*Penicillium commune*	Forward	92	99.56
AS/PDA/5/3/2	Uncultured ascomycete	Reverse	93	98.99
	*Penicillium crustosum*	Reverse	93	98.99
	*Penicillium crustosum*	Reverse	93	98.99
AS/PDA/4/3/2 ^c^	Uncultured *Penicillium*	Forward	71	99.49
	*Talaromyces purpurogenus*	Forward	69	99.47
	Fungal endophyte SPSX01	Forward	69	99.47
AS/PDA/4/3/2	Uncultured *Penicillium*	Reverse	77	99.83
	*Talaromyces purpurogenus*	Reverse	75	99.83
	Fungal endophyte SPSX01	Reverse	75	99.83
AS/PDA/11(2) ^d^	Uncultured *Penicillium*	Forward	71	98.29
	*Talaromyces purpurogenus*	Forward	70	98.25
	Fungal endophyte SPSX01	Forward	70	98.25
AS/PDA/11(2)	Uncultured *Penicillium*	Reverse	88	98.84
	*Talaromyces albobiverticillius*	Reverse	81	98.66
	*Talaromyces purpurogenus*	Reverse	80	92.82

^a^ EU/PDA/ST/11GPC, strain 11GPC isolated from *Euphorbia myrsinites* stem on potato dextrose agar (PDA); ^b^ AS/PDA/5/3/2, strain 5/3/2 isolated from *Alstonia scholaris* leaves in PDA; ^c^ AS/PDA/4/3/2, strain 4/3/2 isolated from *Alstonia scholaris* leaves in PDA; ^d^ AS/PDA/11(2), strain 11(2) isolated from *Alstonia scholaris* leaves in PDA.

**Table 2 metabolites-11-00174-t002:** Primers used for PCR amplification for fungal identification.

Primer’s Name	Sense	Sequence 5′-3′	Gene
V9g	Forward	TTACGTCCCTGCCCTTTGTA	ITS1 and ITS2
ITS4	Reverse	TCCTCCGCTTATTGATATGC	regions + 5.8S

## Data Availability

The data of this study is available in the article and Appendix A.

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
