# Peer review of "HPTLC-Based Chemical Profiling: An Approach to Monitor Plant Metabolic Expansion Caused by Fungal Endophytes"

_metabolites, 2021, doi:10.3390/metabo11030174_

Round 1

Reviewer 1 Report

After reading this second version of the manuscript, I can indicate it has improved substantially. And, although my concerns about not using MS or NMR for comparing the techniques, with the one presented, I can understand the work is focused on the use of this HPTLC technique itself.

From this point of view, I am more in favor of accepting the manuscript with some improvements that I do not know if authors can introduce:

Figure 2a. Still not see the benefit of this figure. Are the colors the same ones as for 2b? attending to the 2a plots there are other groups. Can a third PC be added to the 2a representation? that helps sometimes to see the real closeness of the dots.

Figure 4. There is a typo in the oxisporum. I suggest highlighting the combinations with inhibition. I do not see inhibition in all pairs. Also, the axenic controls of the pathogens in the top and the Endophytes on the left of each row, would be convenient at the same days as in the combinations. are all pairs incubated for the same days? if not, please indicate.

Regards

Author Response

We thank for your valuable commentaries and suggestions which had improved the quality of our work. The comments were attended as follows.

REVIEWER 1

After reading this second version of the manuscript, I can indicate it has improved substantially. And, although my concerns about not using MS or NMR for comparing the techniques, with the one presented, I can understand the work is focused on the use of this HPTLC technique itself.

From this point of view, I am more in favor of accepting the manuscript with some improvements that I do not know if authors can introduce:

Figure 2a. Still not see the benefit of this figure. Are the colors the same ones as for 2b? attending to the 2a plots there are other groups. Can a third PC be added to the 2a representation? that helps sometimes to see the real closeness of the dots.

ANS) Figure 2a indicates the overall chemical differentiation of the diverse fungal strains as reflected on PCA score plot, which is important data for the manuscript. Figure 2b is for the detailed metabolic similarity based on hierarchical analysis with PCS scores. Thus, this image is the proof that chemical information gets it by HPTLC can be well approached by MVDA as in fingerprinting-metabolomics studies. However, if the Editor decide that it is not necessary, we will follow the decision.

A third component was added to the PCA plot, which as the reviewer suggested, it helped to better visualize clusters’ separation.

Figure 4. There is a typo in the oxisporum. I suggest highlighting the combinations with inhibition. I do not see inhibition in all pairs. Also, the axenic controls of the pathogens in the top and the Endophytes on the left of each row, would be convenient at the same days as in the combinations. are all pairs incubated for the same days? if not, please indicate.

ANS) According to the suggestion, Figure 4 was edited and added to the text.

Reviewer 2 Report

Review of Manuscript for the Journal of Metabolites

Manuscript ID: metabolites-1108054

Title: HPTLC-based chemical profiling: An approach to monitor plant metabolic expansion caused by fungal endophytes

In this manuscript, isolating Fungal endophytes from two latex bearing species, the authors use High performance thin layer chromatography (HPTLC) to metabolically profile the samples. Monitoring the interactions among fungal endophytes, the host plant and potential phytopathogens, they study the role of fungal endophytes in plant defense mechanisms either directly or by biotransformation/induction of metabolites.

However, chemical profiling based only on HPTLC data is not enough. HPTLC results needs to be supported by NMR and mass spectroscopy analyses which are reproducible and high precision techniques for metabolomics studies.

Author Response

We thank for your valuable commentary and suggestion. The request was answered as follows.

REVIEWER 2

Title: HPTLC-based chemical profiling: An approach to monitor plant metabolic expansion caused by fungal endophytes

In this manuscript, isolating Fungal endophytes from two latex bearing species, the authors use High performance thin layer chromatography (HPTLC) to metabolically profile the samples. Monitoring the interactions among fungal endophytes, the host plant and potential phytopathogens, they study the role of fungal endophytes in plant defense mechanisms either directly or by biotransformation/induction of metabolites.

However, chemical profiling based only on HPTLC data is not enough. HPTLC results needs to be supported by NMR and mass spectroscopy analyses which are reproducible and high precision techniques for metabolomics studies.

 ANS) The aim of this research was not the identification of specific metabolites, but classification and selection of fungal strains based on their metabolite pools before and after biological interactions. It is rather close to chemical fingerprinting. As reviewer mentioned, currently, there are two popular analytical platforms: NMR and MS. Each of them has own pros and cons regarding sensitivity, broadness of metabolites, signal resolution and so on. Due to their imperfection a proper analytical method should be selected with an optimized protocol: macroscopic- or microscopic view. In particular, one of the advantages of NMR based methods over MS-based ones would be easiness to get chemical characterization, a kind of chemical fingerprinting because the information on chemical structure could be deduced directly from chemical shifts. Besides the popular methods, however, many other analytical platforms such as IR, NIR, UV are being used for the research related to chemical fingerprinting or footprinting (e.g. differentiation coffee, honey, flower colors…). In the context, we would like to apply HPTLC to metabolic profiling works. As mentioned in the introduction, HPTLC is said to have some advantage for the profiling works: a variety of available chemical reaction to visualize diverse groups of metabolites, easiness to isolate target signals to identify target signals, and possibility of parallel analysis. Moreover, we added several references where the only metabolomics tool used is HPTLC, without identification of metabolites, and with identification of metabolites with analysis of specifics HPTLC bands by NMR or MS. We should highlight that neither NMR or MS were used to corroborated HPTLC results because this approach has been previously proven to be enough to produce metabolomics and fingerprinting data. Thus, our aim is to show HPTLC as a good platform to capture the metabolic diversity produced during fungal interactions and correlate it to their biological functions. The Use of other techniques such as NMR or MS was added in the discussion as on-line or off-line analytical methods exclusively used for discriminant metabolites.

In fact, we measured NMR of the samples as a fingerprinting method but due to the low sensitivity, the quality of NMR spectrum could not provide a proper chemical information.

Reviewer 3 Report

Please provide additional references regarding the application of the HPTLC for the metabolomics studies.

The metabolites identification is an essential component of the assays of this kind; therefore the authors should indicate orthogonal techniques which would be suitable to complement the current analytical technique.

In addition, the authors should explain/discuss the reproducibility of Rf values.

Author Response

We thank for your valuable commentaries and suggestions which had improved the quality of our work. The comments were attended as follows.

REVIEWER 3

Please provide additional references regarding the application of the HPTLC for the metabolomics studies.

ANS) The HPTLC application-related references were added to the revised manuscript including different type of analyzed samples and biological questions answered by HPTLC-based metabolomics.

The metabolites identification is an essential component of the assays of this kind; therefore, the authors should indicate orthogonal techniques which would be suitable to complement the current analytical technique.

ANS) The Use of other techniques such as NMR or MS was added in the discussion as on-line or off-line analytical methods used for discriminant metabolites.

In addition, the authors should explain/discuss the reproducibility of Rf values.

ANS) References and discussion about Rf values variation and reproducibility were added.

Round 2

Reviewer 2 Report

Review of Manuscript for the Journal of Metabolites

Manuscript ID: metabolites-1108054

Title: HPTLC-based chemical profiling: An approach to monitor plant metabolic expansion caused by fungal endophytes

I acknowledge the efforts of the authors to improve the manuscript and although I understand the concept of a chemical fingerprinting approach, I still believe that the comparison of the HPTLC analysis results with NMR or/and MS analyses is essential. Therefore I am afraid that I cannot suggest publication. 

This manuscript is a resubmission of an earlier submission. The following is a list of the peer review reports and author responses from that submission.

Round 1

Reviewer 1 Report

I believe it was treated with capability. The article seems too long, there are some editorial errors but overall it is well written. I think it is an iteresting study and a new application of HPTLC.

Reviewer 2 Report

Authors present a HPTLC-based metabolomics for a series of endophytes isolated from two plants. There is an attempt to compare with NMR but complexity of the spectra made them to directly discard NMR. There is no attempt to compare with Mass Spectrometry, the technique of choice for metabolomics nowadays.
Authors perform PCAs and color the samples according to dendrogram from hierarchical cluster. but there is no clear separation in the PCAs of the three clades differentiated in the hierarchical cluster analysis. Authors also describe the PLS analysis and conclude that metabolomic variation is related to colony size. Although there are multiple records that indicate metabolomic variation is correlated clearly with medium components than the phase of growth on agar.
Figure 3 is described as interaction between different strains with growth inhibition zones. Pictures only show interaction of colonies that get to close contact, but no real zones of inhibition are present in the picture.
Conclusion on results from figure 6 are that at 15,8ug/mL the addition of extracts from the plant to the growth medium increases 12 times the generation of antifungal properties in agar. But that seems a situation related to the specific experimental data as other concentrations do not show such a difference. In addition, when comparing with antifungal properties in liquid medium with the addition of plant extract it seems that the effect is the opposite.
In my opinion, this study does fit into the scope of this journal. But it needs to improve comparing results with mass spectroscopy. Also, the amount and conclusions from the data need to be improved. In addition, the presentation/disposition of the HPTLC results is quite confusing. A schema may help in this.

Reviewer 3 Report

Review of Manuscript for the Journal of Metabolites

Manuscript ID: metabolites-1046910

Title: HPTLC-based metabolomics: An approach to monitor the metabolization of plant metabolites by fungal endophytes and contribution to plant defense.

In this manuscript, the authors study the role of fungal endophytes in plant defense mechanisms either directly or by biotransformation/induction of metabolites. Isolating Fungal endophytes from two latex bearing species they use High performance thin layer chromatography (HPTLC) to metabolically profile the samples and to monitor the interactions among fungal endophytes, the host plant and potential phytopathogens. However, I am afraid that the manuscript is not scientifically robust. The authors are based only on HPTLC analysis for their study. Chemical structure elucidation is missing, as the study lacks 1HNMR and HRMS analyses that are fundamental techniques in terms of reproducibility and high precision. HPTLC may serve as a supplementary tool, combined with NMR- or MS-based analyses, but is not enough to be employed for metabolomics studies by itself.